# Digital Reconstructions Using Linear Regression: How Well Can It Estimate Missing Shape Data from Small Damaged Areas?

**DOI:** 10.3390/biology11121741

**Published:** 2022-11-30

**Authors:** Ana Bucchi, Antonietta Del Bove, Sandra López-Lázaro, Fernanda Quevedo-Díaz, Gabriel M. Fonseca

**Affiliations:** 1Centro de Investigación en Odontología Legal y Forense (CIO), Facultad de Odontología, Universidad de La Frontera, Avenida Francisco Salazar 01145, Temuco 4780000, Chile; 2Departament d’Història i Història de l’Art, Universitat Rovira i Virgili, Avinguda de Catalunya 35, 43002 Tarragona, Spain; 3Institut Català de Paleoecologia Humana i Evolució Social (IPHES-CERCA), Zona Educacional 4, Campus Sescelades URV (Edifici W3), 43007 Tarragona, Spain; 4Departamento de Antropología, Facultad de Ciencias Sociales, Universidad de Chile, Santiago 7800284, Chile

**Keywords:** geometric morphometrics, accuracy, cranial reconstruction, craniofacial approximation

## Abstract

**Simple Summary:**

Paleontologists, anthropologists and forensic scientists work with skeletal evidence that is often damaged or fragmented. Inferring what the original morphology of the bones was like is important for reconstructing fossils or identifying individuals. In this paper, we evaluate how accurate a statistical method (linear regression) is for estimating missing shape data. For this purpose, we worked with 3D models of complete human zygomatics (a face bone) that were altered to simulate damage, and reconstructed them using this method. We then evaluated how closely the original morphology resembled the reconstructed one. We conclude that this method can faithfully estimate the original anatomical data, especially when the damage is small, but the error increases significantly with increasing damage size.

**Abstract:**

Skeletal remains analyzed by anthropologists, paleontologists and forensic scientists are usually found fragmented or incomplete. Accurate estimations of the original morphologies are a challenge for which several digital reconstruction methods have been proposed. In this study, the accuracy of reconstructing bones based on multiple linear regression (RM) was tested. A total of 150 digital models from complete zygomatics from recent past populations (European and African American) were studied using high-density geometric morphometrics. Some landmarks (i.e., 2, 3 and 6) were coded as missing to simulate incomplete zygomatics and the missing landmarks were estimated with RM. In the zygomatics, this simulated damage affects a few square centimeters or less. Finally, the predicted and original shape data were compared. The results indicate that the predicted landmark coordinates were significantly different from the original ones, although this difference was less than the difference between the original zygomatic and the mean zygomatic in the sample. The performance of the method was affected by the location and the number of missing landmarks, with decreasing accuracy with increasing damaged area. We conclude that RM can accurately estimate the original appearance of the zygomatics when the damage is small.

## 1. Introduction

Digital methods that recreate the anatomy of incomplete bones are relevant for sciences that often work with damaged or fragmentary materials that cannot be replaced, such as fossils or unidentified persons in forensic sciences [1,2,3,4,5,6,7]. Digital reconstructions are more automatic and reproducible than manual reconstructions, but it is necessary to know the error associated with these virtual methods to know how accurate the reconstructions are.

One of the digital methods that has been proposed as one of the most accurate for reconstructing incomplete bones is RM. The RM shows fewer estimation errors than other commonly used digital methods in paleoanthropology (e.g., thin plate spline) [2,6,8,9] when a large reference sample size is available, and it makes use of more biological information as it considers the patterns of variation and covariation among the landmark positions [10]. In this method, the shape of the bones is studied through geometric morphometrics and the coordinates of each landmark are regressed on all other landmarks from the set of undamaged specimens. Then, the location of the missing landmarks is predicted by the regression model [2].

Although sciences such as forensics, which seeks to identify individuals through facial reconstructions [11,12], or paleontology, which seeks to restore the facial anatomy of fossils [5,13], benefit from methods that achieve accurate reconstructions, the RM is not a widely used method as it requires large reference samples [2,10]. Additionally, despite the promise of this method, its accuracy in realistic scenarios remains to be investigated. To our knowledge, the only study that uses a large sample size and reconstructs human skulls based on a reference sample of the same species estimates missing landmarks that have been randomly selected [10]. This causes the estimated landmarks not to have been connected to each other, which affects the error estimate of the method since the method exploits the fact that physically close landmarks show a redundancy of information and a higher covariation. This implies that the shape data estimation method can be more accurate when estimating disaggregated landmarks than when estimating closely spaced landmarks. The performance of RM within a single species is currently unknown when the bone damage is in one anatomical region. This paper contributes to answer this question by analyzing the predictive power of the method when landmarks are absent in a particular anatomical region of the skull (i.e., zygomatic bone) using a robust reference sample. 

The aim of this study was to test the accuracy of RM for reconstructing zygomatics by simulating damaged bones from an undamaged reference sample (from African American and European populations) and then restoring them using RM. Different scenarios were simulated in which the extent of the damage was varied. We predicted that RM can confidently estimate shape and reduce the uncertainty of estimating missing shape data. We also expected the accuracy to be higher for smaller damaged areas and to vary depending on the sample composition (i.e., Italians vs complete sample), as previous studies have shown that cranial traits vary across populations [14,15,16,17,18], which may act against the performance of the reconstructions.

## 2. Materials and Methods

### 2.1. Sample

The sample was composed of 150 three-dimensional (3D) models from crania obtained via photogrammetry and CT scans (Table 1). The sex of the individuals was known and there were no age data.

The crania belonged to four osteological collections of European or African American origin: (i) the Lynn Copes digital Collection (Black Americans) [19]; (ii) the Olóriz Collection [20]; (iii) the Museum of Anthropology G. Sergi [21] and (iv) the Anthropological Museum of Florence [22]. 

Although the samples consisted of different sexes and populations, the main analyses were completed by pooling the individuals together, as separating by population and sex will affect the sample sizes and the statistical power of the tests. A second reconstruction analysis was carried out exclusively for the Italian subsample, as it was the only collection with a relatively large number of individuals (119). The zygomatics involving trauma or pathologies were excluded from the analysis. Left zygomatics were preferred and their antimeres were reflected when necessary.

**Table 1 biology-11-01741-t001:** Sample analyzed in this study.

Sample Origin	Total	Males	Females	Digitalization Technique	Collection
African Americans	20	9	11	CT scan [19]	Terry collection of the Natural History Museum (Washington, DC, USA)
Spanish	11	8	3	Photogrammetry [23]	Olóriz Collection at the Universidad Complutense de Madrid (Madrid, Spain)
Italian	77	43	34	Photogrammetry [23]	Anthropological Museum of Florence (University of Florence, Florence, Italy)
	42	16	26	Photogrammetry [23]	Museum of Anthropology G. Sergi (Sapienza University of Rome, Rome, Italy)

### 2.2. Data Acquisition

High-dimensional geometric morphometrics, i.e., placing fixed landmarks and semi-landmarks with short distances between them to achieve robust characterizations of morphological variation [24], provide more anatomical information than digitizing more widely spaced landmarks. It thus increases the chances of confidently reconstructing the damaged parts of the bone based on the present shape data. For this reason, on each zygomatic, seven fixed landmarks (Table 2) and 23 surface semi-landmarks were manually digitized onto all specimens using Avizo 8.1.1 (Visualization Sciences Group) (Figure 1). We selected the zygomatic bone because it was one of the best preserved bones of the facial skeleton in our sample.

The data for the 150 specimens presented no missing landmarks and were then imported into R (V 1.4.1106) (R Core Team 2021) using the Arothron package [25]. A generalized Procrustes analysis was carried out to standardize by location, rotation and scale. The semi-landmarks were allowed to slide in order to minimize the bending energy using the R Package Morpho [26].

**Table 2 biology-11-01741-t002:** Fixed landmarks used in this study [27,28].

Number	Landmark	Definition
1	Inferior zygotemporale	Most inferior point in the temporozygomatic suture
2	Superior zygotemporale	Most superior point in the temporozygomatic suture
3	Frontomalare temporale	Most posterior point in the frontozygomatic suture
4	Frontomalare orbitale	Most anterior point in the frontozygomatic suture
5	Zygomaxillare orbitale	Most superior point in the zygomaticomaxillary suture
6	Inferior Zygomaxillare	Most inferior point in the zygomaticomaxillary suture
7	Inferior zygosphenoid^1^	Most inferior point in the zygomaticosphenoid suture, in the orbit

### 2.3. Identifying Outliers

The function *plotOutliers()* (geomorph package [29]) was used to determine whether there were abnormal shape data. Bones that fall away from the mean shape were reviewed to find mistakes during the landmark digitization process, and digitized again.

### 2.4. Simulation Design

The RM method was evaluated for three different case scenarios by varying the number and location of the missing landmarks. In Case 1, two missing landmarks were simulated (e.g., fixed landmarks 1 and 2, Figure 1), which represented an individual with an incomplete zygomatic process. Case 2 simulated a damaged orbit (missing fixed landmarks 4, 5 and 7) and in Case 3, semi-landmarks 16–21 were missing, representing an incomplete zygomatic body (Figure 1). Case 1 represented the smaller damaged area and Case 3 the largest. The simulations were completed by setting the landmark coordinates involved in each case as unknown. This process was carried out in 30 randomly selected zygomatics for each case. Therefore, 90 simulations using RM were performed in total. Each specimen subject in this simulation was labelled as a “target zygomatic”, whereas the remaining 149 specimens were referred to as “reference sample”.

The missing landmark values were estimated by means of RM using the *estimate.missing()* function (geomorph package [29]). At the end of this step, there were two sets of coordinates for the surface points for each target specimen; the original or true coordinates, corresponding to the surface points digitalized in the original zygomatics, and the predicted or reconstructed coordinates for this specimen, after estimating the missing landmarks.

### 2.5. Testing Accuracy within Each Case

The Procrustes distances between the true and predicted shape variables were calculated. 

For comparative purposes, the Procrustes distance was also calculated between the true coordinates of the target zygomatic and the mean zygomatic configuration of the reference sample. A two-sample *t*-test was carried out to compare the Procrustes distances between pairs of zygomatics (reconstructed and original) to the distances between each zygomatic (original) and the mean configuration. This allowed us to examine whether using this method is more accurate than using the mean reference specimen for reconstructing missing parts.

### 2.6. Testing Accuracy across Cases 

To evaluate whether some cases reconstructed the damaged zygomatics more accurately than others, an ANOVA test was carried out to determine if the mean Procrustes distances between the original and predicted zygomatics varied significantly across the cases. A Tukey honestly-significant-difference (HSD) test was then performed to find which specific cases performed the best and worst.

### 2.7. Evaluating the Population Effect

The predictive power of this method was tested on a sample composed of different populations (N = 150, Table 1). To test whether the composition of the sample affects the accuracy of the method for reconstructing missing parts of the zygomatic, a second analysis of the method was performed for the subsample of Italians (N = 119). For the subsample of this population, we ran the same analyses as for the full sample (Section 2.4, Section 2.5, Section 2.6) and then compared the RM accuracy for the full sample and for the subsample of Italians.

## 3. Results

### 3.1. Accuracy for the Reconstruction Method in Each Case

For Case 1 (missing landmarks 1 and 2, Figure 1), the true shape for every zygomatic was more similar to the predictions than to the mean zygomatic bone of the sample (Figure 2). The Procrustes distances between the true and predicted shapes were significantly smaller than the Procrustes distances between the true zygomatic shape and mean zygomatic of the sample (t = −10.21, *p* < 0.01) (Figure 3). 

Cases 2 (missing landmarks 4, 5 and 7) and 3 (missing semi-landmarks 16–21) showed similar results as for Case 1. The predictions were more similar to the true shape than the true shape relative to the mean configuration of the sample in both Cases 2 (t = −13.81, *p* < 0.01) and 3 (t = −8.87, *p* < 0.01) (Figure 2 and Figure 3). 

These analyses for the subsample of Italians behaved similarly to that for the full sample for Case 1 (t = −12.80, *p* < 0.01), Case 2 (t = −13.49, *p* < 0.01), and Case 3 (−9.39, *p* < 0.01).

### 3.2. Accuracy for the Reconstruction Method across all Cases

There were statically significant differences between the means of the Procrustes distances of the original and reconstructed landmark configurations depending on the case (F(2,177) = 43.42, *p* < 0.01) in the complete sample. Reconstructions for Case 1 and Case 2 outperformed Case 3 (*p* < 0.01), whereas Cases 1 and 2 performed similarly (*p* = 0.6) (Table 3).

The results for the subsample of Italians were similar to those obtained for the full sample. For this sample, ANOVA indicated that there were statically significant differences between the means of the Procrustes distances depending on the case (F(2,87) = 58.41, *p* < 0.01) (Table 3). Case 1 and Case 2 showed smaller Procrustes distances between the original and reconstructed landmark configurations (*p* < 0.01) than Case 3, whereas Cases 1 and 2 performed similarly (*p* = 0.9).

## 4. Discussion

We evaluated the performance of RM in missing shape data of damaged zygomatic bones. The results showed that the shape data of the zygomatics reconstructed with this method more closely resembled the original bone than the original zygomatics resembled the average configuration of the sample (Figure 2 and Figure 3). This means that the inter-individual differences were greater than the difference between the original and reconstructed zygomatic through RM. 

There are methods of digital reconstructions that, unlike RM, do not use large sample sizes, but select a geometrically similar individual as a template to reconstruct an incomplete individual [2,7]. The results of our study indicate that estimating missing shape data more accurately requires statistical analysis (RM). A very important result, however, is that the best estimates were for the cases with the least damage (Case 1 and 2) where a few landmarks were missing (2 and 3). These cases performed significantly better than Case 3 where larger damage was simulated. This is consistent with previous findings [10,30], which found that the RM method performed well when the absent landmarks were few and disconnected across the skull and that the number of missing landmarks affect the estimation error. Our study further reports that the decrease in the accuracy in relatively large, damaged areas persists even when using a dense map of anatomical landmarks, although RM remains more accurate than using an average bone as a template (Figure 3). 

In practice, the three cases of our study simulated a very small, damaged area which involved only a few millimeters in the zygomatic. It remains to be evaluated whether RM is capable of accurately predicting the shape variables of even large skull defects and whether it is appropriate for the most common cases encountered by experts in their fields. 

Our results might indicate that not only the number of landmarks but also their location may affect the performance of RM. The location of semi-landmarks 16–21 (missing landmarks in Case 3) might be affected by the variability in the projection of the zygomatic and the malar tubercule across populations [14,31], the morphology of the muscle attachments in the area, including a chewing muscle (masseter muscle) at the zygomatic arch and the maxillary process of the zygomatic, as the masticatory force of this and other chewing muscles affect the insertion morphology [32,33]. It would thus be advisable to consider the nature of the surface one is attempting to reconstruct. 

Previous research recommended that in a large reference sample, RM should be used instead of other digital reconstruction methods [10,34]. The reference sample used by Nesser et al. [10] came from different populations across the world, although this variable was not discussed in their study. It is thus possible that the accuracy in studying a smaller reference sample from a sample specific population will outperform that of a larger and more genetically and geographically diverse population, as several studies have shown that craniofacial traits vary according to geographic distance, climatic conditions and population history (e.g., [16,17]). Our results indicated that the accuracy of the reconstructions with smaller, genetically more homogeneous sample RM was similar to that of the full reference sample (Table 3). This suggests that even a smaller sample size than that proposed by Nesser et al. [10] can be sufficient to achieve robust reconstructions if the RM is applied to a more uniform sample. It also suggests that the analysis of a large sample size comprised of collections from different populations, each with a small number of individuals, can accurately reconstruct the damaged zygomatics (Figure 2, Table 3).

## 5. Conclusions

This study indicates that the RM method is accurate for reconstructing incomplete zygomatics for relatively small damaged areas. In all cases, this statistical method achieved more accurate results than using the average sample configuration as the reference zygomatic for the reconstructions.

The performance of the method increases when the damaged area is smaller and suggests that it might depend on the location of the damage. The population from which the reference sample is taken affects the accuracy of the reconstructions and indicates that it is possible to use a smaller sample size from a more genetically homogeneous sample to obtain results similar to those obtained with a larger and more diverse reference sample.

## Figures and Tables

**Figure 1 biology-11-01741-f001:**
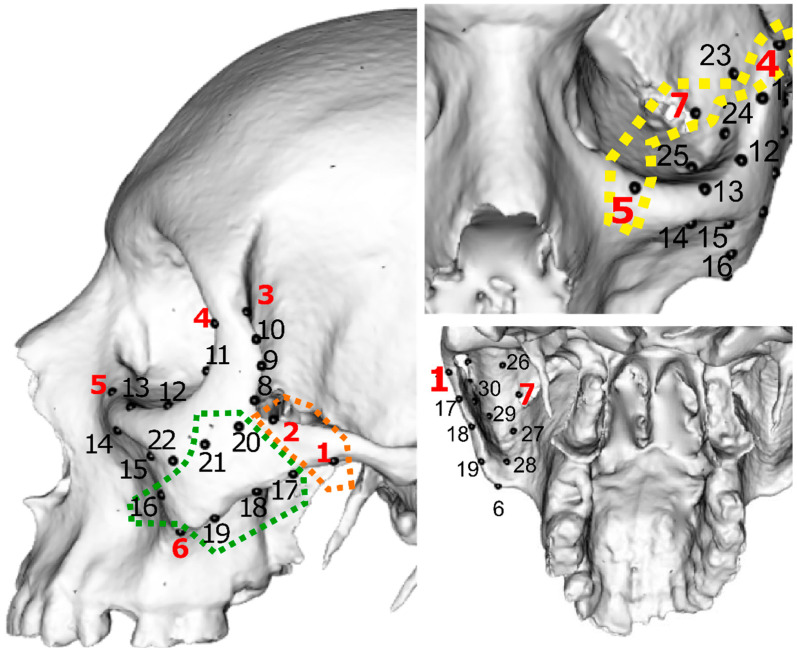
Landmark configuration used in this study. Fixed landmarks are indicated by red numbers (1–7, Table 2) while semi-landmarks are given by black numbers (8–30). Study cases are enclosed with dots: Case 1 (orange), Case 2 (yellow) and Case 3 (green).

**Figure 2 biology-11-01741-f002:**
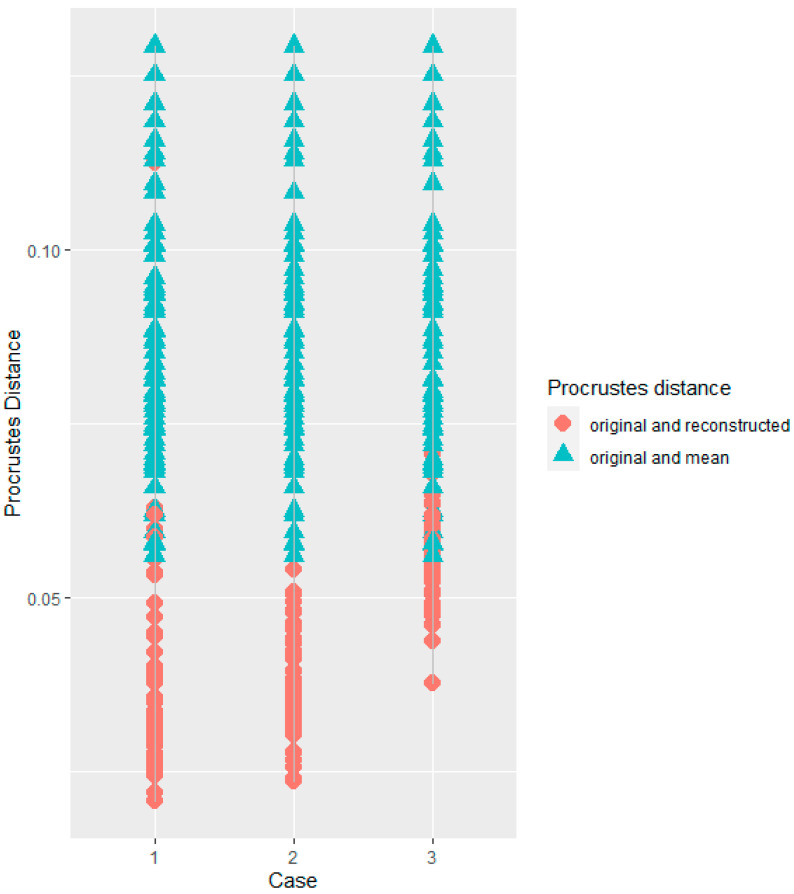
Scatterplot showing the predictive power of the reconstruction method for Cases 1–3 in the complete sample. Red dots indicate the Procrustes distances between the true and predicted landmark coordinates. Light-blue triangles indicate the Procrustes distances between the landmark coordinates of the original zygomatic and the mean configuration of the reference sample.

**Figure 3 biology-11-01741-f003:**
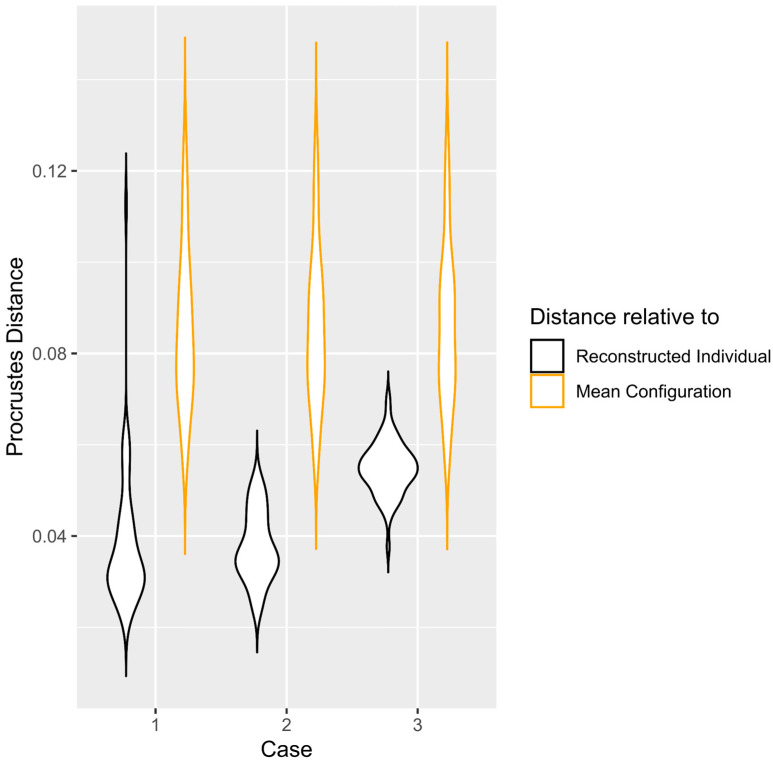
Violin plots showing the Procrustes distances between the original and reconstructed landmark configurations (in black) and between the original and average landmark set (in yellow), for the three cases. The data correspond to the full sample and the subsample of Italians, since the reconstructions in both samples had similar Procrustes distances in relation to the original zygomatics (see Results).

**Table 3 biology-11-01741-t003:** ANOVA post hoc pairwise comparisons (Tukey’s HSD) of the Procrustes distance between the original and the reconstructed landmark configuration and between the original and mean landmark configuration in Cases 1, 2 and 3. *p*-values are in parentheses. Significant results are given in bold.

Procrustes Distance Comparison	Case 1–Case 2	Case 1–Case 3	Case 2–Case 3
Original-Reconstructed Landmark sets in the complete sample	0.002 (0.6)	−0.016 **(*p* < 0.01)**	0.018 **(*p* < 0.01)**
Original-Reconstructed Landmark sets in the Italian sample	0.002 (0.8)	−0.017 **(*p* < 0.01)**	−0.017 **(*p* < 0.01)**
Original-Mean Landmark sets in the complete sample	0.002 (0.8)	−0.008 **(0.02)**	−0.009 **(*p* < 0.01)**
Original-Mean Landmark sets in the Italian sample	−0.001 (0.8)	−0.009 **(*p* < 0.01)**	−0.004 **(*p* < 0.01)**

## Data Availability

Publicly available datasets (3D models) were analyzed in this study. This data can be found here: https://www.lynncopes.com/human-ct-scans.html (accessed on 18 March 2021). The landmarks dataset that support the findings of this study are openly available in Zenodo at https://doi.org/10.5281/zenodo.6857007.

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
