# Peer review of "Digital Reconstructions Using Linear Regression: How Well Can It Estimate Missing Shape Data from Small Damaged Areas?"

_biology, 2022, doi:10.3390/biology11121741_

Round 1

Reviewer 1 Report

This paper presents an assessment of the accuracy of a commonly used statistical reconstruction method for missing 3D data in geometric morphometrics, specifically for when data are missing on areas of differing sizes on the zygomatic bone. Such studies are an important part of the scientific process and thus the contribution to the theoretical literature, while not original, is still worthy. Nevertheless, the aims of the study are not clearly or accurately presented in the manuscript. In addition, the methods are not discussed in sufficient detail, and are frequently unclear and/or inappropriate to the questions being asked. As such, the contribution of the results do not achieve their potential. Furthermore, the discussion and conclusions are highly speculative with an occasional inappropriate tone and lack of contextualisation. I would recommend this paper be revised based on my attached comments and resubmitted. I have split these comments into major and minor concerns, although the number of minor points are quite substantial.

Reviewer 2 Report

The submission is a complete manuscript entitled "Digital reconstructions using linear regression: how well can it estimate missing shape data from small damaged areas?"

After carefully evaluating the manuscript, I noticed weaknesses that should be improved before publication. Here I provide suggestions related to each section. I hope the authors take my comments as constructive advice.

Abstract:

The Abstract describe the work, making it clear what was known before, what was done and what was achieved.

·         In line 29: "Some landmarks are coded" sounds vague. The authors should consider mentioning the number of landmarks coded.

·         l.30: The authors used the acronym "MRI", which isn't used again in the manuscript. Please, check its usage and meaning.

Introduction:

The authors provide a brief background of the work done so far and present the purpose of the study.

·         However, they do not clarify why the interest in the zygomatic and not in any other bone. The author should explain better the reasons for selecting this bone.

·         The text between l.56-65 looks more like something that would fit in a Discussion section and not in the Introduction.

·         l.69-74. It is unclear if the authors are staying the hypotheses in this section. Please, if that's the case, formulate them clearer.

Material and methods:

·         Please, clarify which data was collected using photogrammetry and which using a CT scan.

·         Please provide further explanations about the sample composition in terms of age and sex. If not, clarify why that information hasn't been taken into account.

·         l.94-95. The authors' definition of high-dimensional geometric morphometrics looks simplistic. Please, reformulate.

Results

·         The labels in Figure 2 are difficult to read. Please, correct this.

·         There are inconsistencies in the way "Figures" are called in the text, e.g., "(Figure 2)" page 5, and “(Fig. 2)” page 7.

·         Figure 3 has been captioned as Figure 6 multiple times. Figure 6 doesn't exist in the manuscript.

Discussion:

·         The authors should consider introducing here a critical evaluation of key areas addressed, derive conclusions from the results obtained, and integrate them with those in the literature.

·         At this point, if the authors arrived at a similar conclusion to Neeser et al. 2009, then the novelty of this study is unclear. If that’s not the case, please, reformulate for clarity. Also, if the authors aim to introduce a new methodological approach, I recommend providing more detailed information.

·         Case 2 is not discussed in this section, only 1 and 3.

·         Please correctly reformulate the statement from l.216-217 and l.254-255 with the results obtained so that the real impact and outcomes obtained in the current study are clear to the readers.

Conclusions

This section only has a few lines. It needs to be more elaborated. The authors should refer to the main finding of the research completed, explain the study implications, and suggest further studies (if planned) in more detail.

References

It seems the authors didn't follow a specific citing style, and not always the number used matches the document cited on the list of References: e.g., Neeser et al. is noted as [20] (l.57) and listed as 12.

English language and style

There are inconsistencies in verb forms, unclear sentences that need rephrasing, incorrect use of conjunctions, and wordy phrases. 

Reviewer 3 Report

Dear authors;

I recommend you consider the following questions and corrections:

Table 1. Does the term zigotemporale given among the landmarks exist in Anatomy terminology? If so, it should be supported via a reference? Was “zigotemporale” actually meant to be called “zygomaticotemporale”?

Table 1. Is the term z”igomaxillare orbitale” was used first time in this article?

Table 1. Does the term zigotemporale given among the landmarks exist in Anatomy terminology? If so, it should be supported via a reference?

Table 1. Anatomical inscriptions must be written in accordance with the rules of Nomina Anatomica.

Line 134-  If the actual value for the images of the same individuals is compared with the value resulting from the application, the dependent test should be applied. therefore, it should be clearly stated whether it is an independent sample or a dependent sample.

Thanks to the authors for the nice work.

Round 2

Reviewer 2 Report

The authors have responded to all the suggestions made.